# Myopia prevalence and ocular biometry in children and adolescents at different altitudes: a cross-sectional study in Chongqing and Tibet, China

Yongguo Xiang,[1,2] Hong Cheng,[1,2] Kexin Sun,[1,2] Shijie Zheng,[1,2] Miaomiao Du,[1,2] Ning Gao,[1,2] Tong Zhang,[2] Xin Yang,[1] Jiuyi Xia,[1] Rongxi Huang,[3] Wenjuan Wan,[1,2] Ke Hu  [1,2]

YX and HC contributed equally. WW and KH contributed equally.

For numbered affiliations see end of article.

**Correspondence to**
Ke Hu; 42222@qq.com and Ms Wenjuan Wan; wanwenjuancqums@163.com

## ABSTRACT

**Objective** To investigate the differences in myopia prevalence and ocular biometry in children and adolescents in Chongqing and Tibet, China.

**Design** Cross-sectional study.

**Setting** The study included children and adolescents aged 6–18 years in Chongqing, a low-altitude region, and in Qamdo, a high-altitude region of Tibet.

**Participants** A total of 448 participants in Qamdo, Tibet, and 748 participants in Chongqing were enrolled in this study.

**Methods** All participants underwent uncorrected visual acuity assessment, non-cycloplegic refraction, axial length (AL) measurement, intraocular pressure (IOP) measurement and corneal tomography. And the participants were grouped according to age (6–8, 9–11, 12–14 and 15–18 years group), and altitude of location (primary school students: group A (average altitude: 325 m), group B (average altitude: 2300 m), group C (average altitude: 3250 and 3170 m) and group D (average altitude: 3870 m)).

**Results** There was no statistical difference in mean age (12.09±3.15 vs 12.2±3.10, p=0.549) and sex distribution (males, 50.4% vs 47.6%, p=0.339) between the two groups. The Tibet group presented greater spherical equivalent (SE, −0.63 (−2.00, 0.13) vs −0.88 (−2.88, −0.13), p<0.001), shorter AL (23.45±1.02 vs 23.92±1.19, p<0.001), lower prevalence of myopia (39.7% vs 47.6%, p=0.008) and flatter mean curvature power of the cornea (Km, 43.06±1.4 vs 43.26±1.36, p=0.014) than the Chongqing group. Further analysis based on age subgroups revealed that the Tibet group had a lower prevalence of myopia and higher SE in the 12–14, and 15–18 years old groups, shorter AL in the 9–11, 12–14 and 15–18 years old groups, and lower AL to corneal radius of curvature ratio (AL/CR) in all age subgroups compared with the Chongqing group, while Km was similar between the two groups in each age subgroup. Simple linear regression analysis showed that SE decreased with age in both the Tibet and Chongqing groups, with the Tibet group exhibiting a slower rate of decrease (p<0.001). AL and AL/CR increased with age in both the Tibet and Chongqing groups, but the rate of increase was slower in the Tibet group (p<0.001 of both). Multiple linear regression analysis revealed that AL had the

## STRENGTHS AND LIMITATIONS OF THIS STUDY

⇒ This study investigated the differences in myopia prevalence and ocular biometry in children and adolescents residing at different altitudes in China.

⇒ The limited sample size employed in this study may hinder a comprehensive evaluation of the ocular biometric characteristics in children and adolescents.

⇒ The vigorous accommodative system in children may introduce potential inaccuracies in non-cycloplegic refraction, potentially resulting in an overestimation of myopia.

greatest effect on SE in both groups, followed by Km. In addition, the children and adolescents in Tibet presented thinner corneal thickness (CCT, p<0.001), smaller white to white distance (WTW, p<0.001), lower IOP (p<0.001) and deeper anterior chamber depth (ACD, p=0.015) than in Chongqing. Comparison of altitude subgroups showed that the prevalence of myopia (p=0.002), SE (p=0.031), AL (p=0.001) and AL/CR (p<0.001) of children at different altitudes was statistically different but the Km (p=0.189) were similar. The highest altitude, Tengchen County, exhibited the lowest prevalence of myopia and greatest SE among children, and the mean AL also decreased with increasing altitude.

**Conclusions** Myopia prevalence in Tibet was comparable with that in Chongqing for students aged 6–8 and 9–11 years but was lower and myopia progressed more slowly for students aged 12–14 and 15–18 years than in Chongqing, and AL was the main contributor for this difference, which may be related to higher ultraviolet radiation exposure and lower IOP in children and adolescents at high altitude in Tibet. Differences in AL and AL/CR between Tibet and Chongqing children and adolescents manifested earlier than in SE, underscoring the importance of AL measurement in myopia screening.

## BACKGROUND

It is reported that approximately 400 million people worldwide live at altitudes above 1500 m.[1] The Qinghai-Tibet Plateau is the highest geographical unit in the world. Qamdo is

located in the eastern part of the Qinghai-Tibet Plateau, with an average altitude of more than 3500 m. It is characterised by low atmospheric pressure, hypoxia, cold and dry weather and strong ultraviolet radiation (UVR).[2]

Previous study have shown that UVR intensity increases by 10%–12% for every 1000 m of altitude. And approximately 90%–99% of the solar UVR energy that reaches the earth's surface is UVA (315–400 nm wavelength), whereas only 1%–10% is UVB (280–315 nm wavelength).[3] The cornea sits at the anterior aspect of the eye and, like the skin, is highly exposed to UVR. Torii *et al*[4] found that violet light (VL, 360–400 nm wavelength) suppresses myopia progression by upregulating the myopia suppressive gene *EGR1*. Kobashi *et al*[5] reported that there is a small amount of riboflavin in the human corneal stroma, and the ex vivo corneal stiffness significantly increases after VL (375 nm wavelength) irradiation based on physiological riboflavin. Epidemiological data show that the prevalence of keratoconus, pterygium, dry eye disease, cataract and strabismus is significantly higher in the highlands than in the lowlands[2 6–9] and the cornea of the highlanders is thinner than that of the lowlanders.[10 11] Our previous study also found that the prevalence of myopia among children and adolescents in Tibet was lower than that in the plains.[12] However, it is unclear whether ocular biometry is associated with myopia prevalence between high and low altitude populations with different levels of UVR exposure.

The axial length (AL) and AL to corneal radius of curvature ratio (AL/CR) have been reported to be strongly correlated with ocular refraction, and play an important role in the screening and monitoring of myopia.[13] Cycloplegic refraction is considered the gold standard for measuring refractive errors, but limited by the fact that mydriatic drops can induce mydriasis and cycloplegia.[14] Consequently, non-cycloplegic refraction remains a dependable evaluation index in large-scale myopia investigations.

In this study, we compared the prevalence of myopia and ocular biometry in children and adolescents in Chongqing and Qamdo, Tibet, and attempted to explore the effect of altitude on ocular biometry.

## MATERIALS AND METHODS

This school-based cross-sectional study was conducted among children and adolescents aged 6–18 years in Chongqing, a low-altitude region, and in Qamdo, a high-altitude region of Tibet. The Ethics Committee of the First Affiliated Hospital of Chongqing Medical University approved this study and it was conducted in accordance with the tenets of the Declaration of Helsinki. At least one parent or legal guardian of each participant was informed about the study and signed an informed consent form.

### Study population

This study selected Karo district, Tengchen county, Markam county and Drayab county in Qamdo, Tibet, as well as Shapingba district and Banan district in Chongqing, as the investigation sites. A stratified sampling method was employed to select students ranging from the first grade through the senior second level at each investigation site. Ultimately, a total of 480 students from Qamdo, Tibet, and 944 students from Chongqing were included in this school-based study. The specifics of the sample inclusion are presented in table 1.

### Methods

All participants underwent a standardised ophthalmic examination. Uncorrected visual acuity (UCVA) was assessed and recorded in LogMAR scores by using the standard logarithmic visual acuity chart at 5 m. Non-cycloplegic refraction was performed with an

**Table 1** Distribution of sample size and average altitude of each survey site

| Groups | School | | Sample size (student) | Average altitude (m) |
|---|---|---|---|---|
| Tibet | Karo District | Qamdo No. 1 Primary School | 60 | 3250 |
| | | Qamdo Junior High School | 30 | |
| | | Qamdo No. 1 Senior High School | 120 | |
| | Drayab County | Drayab County No. 1 Primary School | 60 | 3170 |
| | | Drayab County Junior High School | 30 | |
| | Tengchen County | Xiexiong Township Central Primary School | 60 | 3870 |
| | | Tengchen County Junior High School | 30 | |
| | Markam County | Qucainka Township Central Primary School | 60 | 2300 |
| | | Mangkang County Junior High School | 30 | 3865 |
| | Total | – | 480 | – |
| Chongqing | Shapingba District | Fenghuang Experimental School | 746 | 319 |
| | Banan District | Banan Middle School | 198 | 331 |
| | Total | – | 944 | – |

autorefractor (Supore, China). AL was measured with the AL-scan (NIDEK, Japan). Intraocular pressure (IOP) was assessed with a non-contact tonometer (NIDEK, Japan). Corneal and anterior chamber parameters were measured with the Pentacam AXL (Oculus Optikgeräte GmbH, Wetzlar, Germany). The quality specification section on the output graph was used to assess the quality, with an 'OK' reading indicating an acceptable quality. All examinations were performed by skilled ophthalmologists. At least three measurements were made for each eye and averaged, but only one eye of each participant was randomly selected for statistical analysis. SE equals diopter (D) of spherical power plus 1/2 diopter (D) of cylindrical power. Myopia was defined as SE <−0.5D and UCVA>0 logMAR.

The investigated indices were as follows: UCVA, SE, AL, IOP and AL to corneal radius of curvature ratio (AL/CR); mean curvature power of the cornea within the central 3 mm circle (km), astigmatism of the front surface (Astig. F); astigmatism of the back surface (Astig. B); corneal thickness at the apex (CCT); white to white distance (WTW); internal anterior chamber depth (ACD) and ACD/AL.

The participants were grouped according to age (6–8 years group, 9–11 years group, 12–14 years group and 15–18 years group), and altitude of location (primary school students: group A (Shapingba and Banan district, average altitude: 325 m), group B (Qucainka township, average altitude: 2300 m), group C (Karo district and Drayab county, average altitude: 3250 and 3170 m, respectively) and group D (Xiexiong Township, average altitude: 3870 m)).

### Patient and public involvement statement

Patients or the public were not involved in the design, or conduct, or reporting or dissemination plans of our research

### Statistical analysis

The statistical analysis was performed with SPSS V.26 (IBM Corp. in Armonk, NY, USA). One-sample Kolmogorov–Smirnov tests and the normal distribution histogram method were used to assess the normality of the distributions of the continuous variables. Independent samples t-test and analysis of variance were employed to compare normally and approximately normally distributed continuous variables, and Mann-Whitney U test and Kruskal-Wallis H test were used to compare heavily skewed continuous variables. Dunn-Bonferroni post-hoc test was conducted for pairwise comparisons. Analysis of covariance test was applied to compare IOP, WTW and ACD between the Tibet and Chongqing groups, with adjustments made for CCT or AL. The $\chi^2$ test was utilised to compare categorical variables. Simple linear regression analysis was performed to examine the relationship between SE, AL, AL/CR and age. Multiple linear regression analysis was conducted to explore the relationship between AL, Km and SE. The significance level was set at $p<0.05$.

### RESULTS

Thirty-two students from Tibet and 181 students from Chongqing were excluded due to lack of qualified ocular biometric results, and 15 students from Chongqing were excluded for wearing corneal contact lenses. Finally, a total of 448 students in Tibet and 748 students in Chongqing were enrolled in this study. The distribution of participants by gender and age is presented in table 2. And there were no statistical difference observed in the mean age (12.09±3.15 vs 12.2±3.10, p=0.549, table 3) and

**Table 2** Distribution of participants of different ages in Tibet and Chongqing groups

| Age (year) | Tibet group | | | Chongqing group | | |
|---|---|---|---|---|---|---|
| | Total | Males | Females | Total | Males | Females |
| 6 | 22 | 10 | 12 | 29 | 15 | 14 |
| 7 | 38 | 17 | 21 | 59 | 24 | 35 |
| 8 | 36 | 20 | 16 | 69 | 39 | 30 |
| 9 | 32 | 20 | 12 | 55 | 22 | 33 |
| 10 | 39 | 19 | 20 | 53 | 34 | 19 |
| 11 | 42 | 19 | 23 | 78 | 39 | 39 |
| 12 | 53 | 34 | 19 | 86 | 40 | 46 |
| 13 | 49 | 18 | 31 | 79 | 35 | 44 |
| 14 | 44 | 25 | 19 | 74 | 49 | 25 |
| 15 | 38 | 20 | 18 | 66 | 35 | 31 |
| 16 | 28 | 11 | 17 | 61 | 13 | 48 |
| 17 | 19 | 10 | 9 | 37 | 9 | 28 |
| 18 | 8 | 3 | 5 | 2 | 2 | 0 |
| Total | 448 | 226 | 222 | 748 | 356 | 392 |

**Table 3** Comparison of demographic data, myopia prevalence and ocular biometric parameters in Tibetan and Chongqing children and adolescents

| | Age (year) | Tibet group | Chongqing group | P value | Adjusted P value* |
|---|---|---|---|---|---|
| Sex: males (%) | Total | 226/448 (50.4%) | 356/748 (47.6%) | 0.339 | |
| Age (year) | Total | 12.09±3.15 | 12.20±3.10 | 0.549 | |
| Myopia prevalence: Number of myopes/Total (%) | 6–8 | 11/**96** (11.5%) | 14/**157** (8.9%) | 0.511 | |
| | 9–11 | 39/**113** (34.5%) | 55/**186** (29.6%) | 0.372 | |
| | 12–14 | 69/**146** (47.3%) | 141/**239** (59%) | 0.025 | |
| | 15–18 | 59/**93** (63.4%) | 146/**166** (88%) | <0.001 | |
| | Total | 178/**448** (39.7%) | 356/**748** (47.6%) | 0.008 | |
| SE (D) | 6–8 | 0 (−0.50, 0.38) | −0.13 (−0.50, 0.25) | 0.191 | |
| | 9–11 | −0.38 (−1.25, 0.13) | −0.44 (−1.13, 0) | 0.829 | |
| | 12–14 | −0.88 (−2.37, 0) | −1.5 (−3.13, −0.50) | 0.001 | |
| | 15–18 | −1.75 (−3.38, −0.88) | −3.63 (−4.88, −2.38) | <0.001 | |
| | Total | −0.63 (−2.00, 0.13) | −0.88 (−2.88, −0.13) | <0.001 | |
| AL (mm) | 6–8 | 22.81±0.76 | 22.94±0.86 | 0.228 | |
| | 9–11 | 23.28±0.85 | 23.57±0.90 | 0.007 | |
| | 12–14 | 23.58±1.01 | 24.17±1.09 | <0.001 | |
| | 15–18 | 24.13±1.03 | 24.86±1.05 | <0.001 | |
| | Total | 23.45±1.02 | 23.92±1.19 | <0.001 | |
| Km (D) | 6–8 | 42.89±1.49 | 43.24±1.40 | 0.067 | |
| | 9–11 | 43.04±1.34 | 43.17±1.27 | 0.398 | |
| | 12–14 | 43.14±1.34 | 43.30±1.45 | 0.302 | |
| | 15–18 | 43.11±1.49 | 43.32±1.32 | 0.398 | |
| | Total | 43.06±1.40 | 43.26±1.36 | 0.014 | |
| AL/CR | 6–8 | 2.90±0.08 | 2.94±0.09 | <0.001 | |
| | 9–11 | 2.97±0.10 | 3.01±0.10 | <0.001 | |
| | 12–14 | 3.01±0.12 | 3.10±0.14 | <0.001 | |
| | 15–18 | 3.08±0.13 | 3.19±0.13 | <0.001 | |
| | Total | 2.99±0.13 | 3.06±0.15 | <0.001 | |
| ACD/AL | 6–8 | 0.132±0.009 | 0.128±0.010 | 0.003 | |
| | 9–11 | 0.136±0.009 | 0.130±0.009 | <0.001 | |
| | 12–14 | 0.135±0.009 | 0.131±0.010 | <0.001 | |
| | 15–18 | 0.134±0.009 | 0.130±0.009 | <0.001 | |
| | Total | 0.134±0.009 | 0.130±0.010 | <0.001 | |
| CCT (µm) | 6–8 | 541.35±32.45 | 545.72±30.08 | 0.278 | |
| | 9–11 | 540.26±27.32 | 552.67±32.32 | 0.001 | |
| | 12–14 | 530.19±30.83 | 551.84±29.48 | <0.001 | |
| | 15–18 | 533.37±28.51 | 548.43±31.35 | <0.001 | |
| | Total | 535.78±30.16 | 550.01±30.80 | <0.001 | |
| WTW (mm) | 6–8 | 11.68±0.40 | 11.75±0.36 | 0.156 | 0.328 |
| | 9–11 | 11.65±0.35 | 11.73±0.35 | 0.071 | 0.771 |
| | 12–14 | 11.57±0.39 | 11.70±0.38 | 0.002 | 0.079 |
| | 15–18 | 11.63±0.36 | 11.69±0.39 | 0.224 | 0.437 |
| | Total | 11.62±0.38 | 11.71±0.37 | <0.001 | 0.039 |
| ACD (mm) | 6–8 | 3.01±0.22 | 2.94±0.26 | 0.031 | 0.003 |
| | 9–11 | 3.16±0.25 | 3.06±0.27 | 0.003 | <0.001 |

Continued

| | Age (year) | Tibet group | Chongqing group | P value | Adjusted P value* |
|---|---|---|---|---|---|
| **Table 3** Continued | | | | | |
| | 12–14 | 3.18±0.26 | 3.18±0.27 | 0.775 | 0.001 |
| | 15–18 | 3.25±0.26 | 3.24±0.27 | 0.867 | <0.001 |
| | Total | 3.15±0.26 | 3.11±0.29 | 0.015 | <0.001 |
| IOP (mm Hg) | 6–8 | 15.30±2.64 | 16.52±3.28 | 0.001 | 0.004 |
| | 9–11 | 14.94±3.05 | 16.52±2.88 | <0.001 | 0.001 |
| | 12–14 | 14.23±3.20 | 16.61±2.83 | <0.001 | <0.001 |
| | 15–18 | 14.01±2.59 | 16.01±2.87 | <0.001 | <0.001 |
| | Total | 14.59±2.96 | 16.43±2.95 | <0.001 | <0.001 |
| Astig. F (D) | Total | 1.05 (0.70, 1.40) | 1.10 (0.80, 1.60) | 0.036 | |
| Astig. B (D) | Total | 0.30 (0.30, 0.40) | 0.30 (0.30, 0.40) | 0.326 | |

*Adjusted AL (WTW, ACD) or CCT (IOP).

ACD, internal anterior chamber depth; AL, axial length; AL/CR, AL to corneal radius of curvature ratio; Astig. B, astigmatism of the back surface; Astig. F, astigmatism of the front surface; CCT, corneal thickness at the apex; D, diopter; IOP, intraoccular pressure; Km, mean curvature power of the cornea within the central 3-mm circle; SE, spherical equivalent; WTW, white to white distance.

sex distribution (males, 50.4% vs 47.6%, p=0.339, table 3) between the two groups.

## Comparison between the Tibet group and the Chongqing group

As shown in table 3, the Tibet group presented a lower prevalence of myopia (39.7% vs 47.6%, p=0.008), greater SE (−0.63 (−2.00, 0.13) vs −0.88 (−2.88, −0.13), p<0.001), shorter AL (23.45±1.02 vs 23.92±1.19, p<0.001), flatter Km (43.06±1.4 vs 43.26±1.36, p=0.014), smaller anterior corneal surface astigmatism (1.05 (0.70, 1.40) vs 1.10 (0.80, 1.60), p=0.036) and smaller AL/CR (2.99±0.13 vs 3.06±0.15, p<0.001) than the Chongqing group, while the astigmatism of the posterior corneal surface was similar between the two groups (0.30 (0.30, 0.40) vs 0.30 (0.30, 0.40), p=0.326). Further analysis based on age subgroups revealed that the Tibet group had a lower prevalence of myopia and higher SE in the 12–14, and 15–18 years old groups, shorter AL in the 9–11, 12–14 and 15–18 years old groups and lower AL/CR in all age subgroups compared with the Chongqing group, while Km was similar between the two groups in each age subgroup (table 3, online supplemental figure S1a–e).

The children and adolescents in Tibet exhibited thinner CCT (535.78±30.16 vs 550.01±30.8, p<0.001), smaller WTW (11.62±0.38 vs 11.71±0.37, p<0.001, after adjustment for AL, p=0. 039), deeper ACD (3.15±0.26 vs 3.11±0.29, p=0.015, after adjustment for AL, p<0.001), lower IOP (14.59±2.96 vs 16.43±2.95, p<0.001, after adjustment for CCT, p<0.001) and greater ACD/AL (0.134±0.009 vs 0.130±0.010, p<0.001) than those in Chongqing. Subgroup analysis based on age revealed similar differences in CCT, IOP and ACD/AL (table 3, online supplemental figure S1f–h). Furthermore, after adjusting for AL, the ACD was found to be deeper in the Tibet group across all age subgroups, while the WTW remained similar.

## Regression analysis

The simple linear regression analysis showed that SE decreased with age in both the Tibet ($R^2$=0.140, p<0.001) and Chongqing ($R^2$=0.352, p<0.001) groups, with the Tibet group exhibiting a slower rate of decrease (B: −0.217 vs −0.393, p<0.001, online supplemental figure S2a). Additionally, AL and AL/CR increased with age in both Tibet (AL: $R^2$=0.192, p<0.001; AL/CR: $R^2$=0.253, p<0.001) and Chongqing (AL: $R^2$=0.335, p<0.001; AL/CR: $R^2$=0.391, p<0.001) groups, but the rate of increase was slower in the Tibet group (B: AL: 0.143 vs 0.223, p<0.001, online supplemental figure S2b; AL/CR: 0.02 vs 0.03, p<0.001, online supplemental figure S2c). Conversely, there was no linear relationship between Km and age for both the Tibet ($R^2$=0.004, p=0.173) and Chongqing ($R^2$=0.001, p=0.377, online supplemental figure S2d) groups. Furthermore, the multiple linear regression analysis revealed that AL had the greatest effect on SE in both groups, followed by Km (table 4).

## Comparison of children among altitude subgroups

As shown in table 5, there was no statistical difference in the mean age (p=0.557) and sex distribution (p=0.979) among the altitude subgroups. Comparison of altitude subgroups showed that the prevalence of myopia (p=0.002), SE (p=0.031, online supplemental figure S3a), AL (p=0.001, online supplemental figure S3b), AL/CR (p<0.001, online supplemental figure S3c) and ACD/AL (p<0.001, online supplemental figure S3d) of children at different altitudes was statistically different, but the Km (p=0.189) were similar. Notably, the highest altitude, Tengchen County, exhibited the lowest prevalence of myopia among children, and the mean AL value also decreased with increasing altitude.

## DISCUSSION

In this school-based cross-sectional study, we conducted a comparison of the prevalence of myopia and ocular

**Table 4** The results of multiple linear regression analysis between SE and AL and Km

| Group | | AL | | | Km | | | Adjusted R$^2$ |
|---|---|---|---|---|---|---|---|---|
| | | B | Sta.β | P value | B | Sta.β | P value | |
| Tibet group (n=448) | SE | −1.567 | −0.880 | <0.001 | −0.649 | −0.500 | <0.001 | 0.651 |
| Chongqing group (n=748) | SE | −1.616 | −0.940 | <0.001 | −0.768 | −0.510 | <0.001 | 0.777 |
| Total (n=1196) | SE | −1.551 | −0.905 | <0.001 | −0.696 | −0.486 | <0.001 | 0.728 |

AL, axial length; B, unstandardised coefficient B; Km, mean curvature power of the cornea within the central 3-mm circle; Sta.β, standardised coefficient beta.

biometry among students aged 6–18 years in Chongqing and Qamdo, Tibet. Our findings indicate that the prevalence of myopia and SE was comparable between the two locations during the age groups of 6–8 years and 9–11 years. However, we observed a lower prevalence of myopia and a higher SE in Tibet compared with Chongqing among students aged 12–14 years and 15–18 years. This discrepancy may be attributed to the slower progression of myopia with advancing age in the Tibetan population. The Tibetan students exhibited a shorter AL within the 9–11, 12–14 and 15–18 years old age groups, as well as

lower AL/CR across all age subgroups in comparison to the students from Chongqing. However, the Km values were found to be similar between the two locations within each respective age subgroup.

Myopia is one of the most common eye disorders globally, caused by both environmental and genetic risk factors.[15] And the increasing prevalence of myopia can be largely explained by increased educational pressures and reductions in the amount of time that children spend outdoors.[16] Qian *et al*[17] and Wang *et al*[12] have reported that Tibetan children and adolescents at high altitudes

**Table 5** Comparison of ocular biometric parameters in altitude subgroups of primary school students

| Parameters | Group A, n=398 | Group B, n=58 | Group C, n=121 | Group D, n=60 | P value |
|---|---|---|---|---|---|
| Sex: males (%) | 201 (50.5%) | 31 (53.4%) | 62 (51.2%) | 31 (51.7%) | 0.979 |
| Age (year) | 9.75±1.85 | 9.87±1.98 | 9.51±2.03 | 9.83±1.78 | 0.557 |
| Myopia prevalence | 99 (24.9%) | 16 (27.6%) | 39 (32.2%) | 4 (6.7%) | 0.002 (A,D: <0.05; B,D: <0.05; C,D: <0.05) |
| SE (D) | −0.38 (−0.88, 0.13) | −0.19 (−0.88, 0.25) | −0.25 (−1.25, 0.25) | −0.06 (−0.50, 0.25) | 0.031 (A,D: 0.019) |
| AL (mm) | 23.39±0.98 | 23.19±0.84 | 23.11±0.90 | 22.97±0.79 | 0.001 (A,C: 0.026; A,D: 0.007) |
| IOP (mm Hg) | 16.57±3.05 | 16.12±2.50 | 14.45±3.05 | 15.17±2.86 | <0.001 (A,C: <0.001; A,D: 0.004; B,C: 0.004) |
| AL/CR | 2.99±0.11 | 2.94±0.09 | 2.95±0.11 | 2.92±0.07 | <0.001 (A,B: 0.005; A,C: 0.001; A,D: <0.001) |
| ACD/AL | 0.130±0.01 | 0.136±0.009 | 0.133±0.009 | 0.134±0.009 | <0.001 (A,B<0.001; A,C: 0.001; A,D: 0.015) |
| Km (D) | 43.22±1.34 | 42.87±1.29 | 43.11±1.46 | 42.93±1.59 | 0.189 |
| Astig. F (D) | 1.10 (0.70, 1.50) | 1.05 (0.70, 1.20) | 1.20 (0.80, 1.60) | 0.95 (0.80, 1.25) | 0.041 |
| Astig. B (D) | 0.30 (0.20, 0.40) | 0.30 (0.30, 0.40) | 0.40 (0.30, 0.40) | 0.30 (0.30, 0.40) | 0.357 |
| CCT (µm) | 550.11±31.2 | 533.17±30.12 | 545.77±28.13 | 532.4±32.50 | < 0.001 (A,B: 0.001; A,D: <0.001; C,D: 0.036) |
| WTW (mm) | 11.73±0.36 | 11.63±0.36 | 11.63±0.38 | 11.71±0.42 | 0.024 (A,C: 0.048) |
| ACD (mm) | 3.03±0.27 | 3.16±0.27 | 3.08±0.24 | 3.07±0.25 | 0.004 (A,B: 0.004) |

ACD, internal anterior chamber depth; AL, axial length; AL/CR, AL to corneal radius of curvature ratio; Astig. B, astigmatism of the back surface; Astig. F, astigmatism of the front surface; CCT, corneal thickness at the apex; D, diopter; IOP, intraoccular pressure; Km, mean curvature power of the cornea; SE, spherical equivalent; WTW, white to white distance.

had a lower prevalence of myopia than their counterparts at low altitudes. In line with their findings, our study also revealed that children and adolescents in Tibet presented a lower overall prevalence of myopia, shorter AL, higher SE and slower myopia progression in comparison to individuals in Chongqing. Furthermore, the prevalence of myopia among children and adolescents in Tibet was found to be lower than previous reports in other plain regions of China.[18–20] Interestingly, the prevalence of myopia and SE of children aged 6–8 and 9–11 years in Tibet in our study was similar to those in Chongqing, and similar findings were reported by Wu *et al*,[21] suggesting that myopia is widespread among children in high altitude regions and that myopia prevention and control in high altitude areas of China are equally challenging.

In recent years, much research evidence suggests that outdoor activity in bright light conditions is a protective factor against myopia.[15 16] The mechanism for this protective effect appears to be related to an increased release of dopamine stimulated by visible light (400–700 nm wavelength), which inhibits increased axial elongation.[22] In addition, Tsubota's team reported that VL, with wavelengths from 360 to 400 nm, can effectively suppress myopia progression in chicks, mice and humans.[4 23 24] Qamdo is located in the eastern part of the Qinghai-Tibet Plateau, with an average altitude of more than 3500 m, and the UVR intensity, especially UVA intensity (315–400 nm wavelength), is stronger in Qamdo than that in Chongqing due to the increase in UVR intensity with altitude.[3] Additionally, Qamdo experiences an average daily sunshine duration of 7.18 hours, approximately twice the duration observed in Chongqing, which amounts to 3.54 hours https://www.weather2visit.com/). The results of our study indicated that children and adolescents in Tibet exhibited shorter AL and slower AL elongation in comparison to their counterparts in Chongqing. Furthermore, our analysis of altitude subgroups revealed a decrease in the mean value of AL with increasing altitude. Although information regarding outdoor activities was not collected in this study for both groups, our previous research demonstrated that children and adolescents in Tibet spend more time outdoors than those in Chongqing,[12] which may be synergistic with stronger UVR intensity and longer average daily sunshine in Tibet in slowing myopia progression.

The findings from the multiple linear regression analysis indicated that the AL and Km were the primary determinants of SE. Notably, the Km remained consistent with age in both Tibet and Chongqing children and adolescents. Consequently, the AL emerged as the primary factor contributing to the disparity in myopia progression between the two locations. Moreover, the differences in AL between Tibet and Chongqing children and adolescents manifested earlier than those in SE, underscoring the importance of AL measurement in myopia screening. Corneal curvature as well as ACD also contribute to myopic progression.[25] In the case of children and adolescents in Tibet, the mean corneal curvature was found to be

flatter compared with those in Chongqing, although with a minimal impact on the difference in myopia. However, after adjusting for AL, it was observed that the ACD in Tibetan children and adolescents was deeper than that in Chongqing. This suggested that the lens in Tibetan individuals may be either flatter or positioned more posteriorly, resulting in a stronger compensatory effect on the elongation of AL.[26 27] However, regrettably, the absence of lens curvature data is attributed to the constraints of the examination equipment, and the investigation into the association between lens curvature and myopia will be undertaken in subsequent research.

The mechanism of myopia development was considered to be associated with scleral matrix remodelling.[28] And scleral matrix remodelling has been shown to contribute to the biomechanical susceptibility of the sclera to accommodation-induced IOP fluctuations, resulting in reduced scleral thickness, AL elongation and axial myopia. The rise in IOP can increase the burden of scleral stretching and cause axial lengthening.[29] In addition, lowering IOP was also considered a potential approach for controlling the progression of high myopia.[30] Several clinical studies have examined the association between IOP and the progression of myopia, but the results were inconsistent.[29] In our study, we observed that IOP levels were consistently higher in children and adolescents residing in Chongqing compared with those in Tibet, across all age subgroups. In a cross-sectional, multicentre, population-based study including 284 937 participants conducted by Liu *et al*,[31] they also reported that IOP was significantly higher in low-altitude populations than in high-altitude populations. This disparity persisted after adjusting for CCT, which may suggest a potential link between elevated IOP and myopia. However, further longitudinal studies are required to establish a causal relationship between IOP and AL elongation. The underlying reasons for the observed disparity in IOP between these two populations remain unclear.

In our study, it was observed that Tibetan children and adolescents exhibited a thinner corneal thickness compared with their counterparts in Chongqing. This finding aligns with previous research conducted by Amit *et al*[10] and Patyal *et al*,[11] who also reported a thinner CCT in individuals residing at higher altitudes. Additionally, our study revealed that the CCT of Tibetan children and adolescents was thinner than what was previously observed in studies conducted at lower altitudes.[32–34] This discrepancy may be attributed to variations in UVR exposure levels, as evidenced by the study conducted by Asharlous *et al*,[35] which found that welders, who are exposed to long-term UVR, exhibited a significantly thinner cornea compared with healthy controls. Hence, it is plausible to consider long-term UVR exposure as a potential contributor to corneal thinning.

Our current study had several limitations. First, the limited sample size employed in this study may hinder a comprehensive evaluation of the ocular biometric characteristics in children and adolescents residing in the

plateau region. Second, the accommodative system of children is vigorous, which may introduce inaccuracies in non-cycloplegic refraction, potentially resulting in an overestimation of myopia. Finally, this study primarily examined variations in ocular biometry among children and adolescents living at different altitudes. However, further research is needed to explore the potential correlation between ocular biometry and behavioural habits. In addition, cross-sectional studies cannot accurately assess the progression of myopia. In the future, we will conduct prospective studies to provide more reliable conclusions.

In conclusion, this study investigated the prevalence of myopia and ocular biometry of children and adolescents aged 6–18 years in Chongqing and Qamdo, Tibet. Students aged 12–18 years had a lower prevalence of myopia and greater SE in Tibet than in Chongqing, but no discernible difference was observed among students aged 6–11 years, indicating a comparatively slower progression of myopia in Tibetan students, which may be related to higher UVR exposure, flatter corneas and lower IOP in Tibetan children and adolescents. In addition, we found that the Tibetan population had thinner corneas than the Chongqing population, which may be associated with long-term UVR exposure.

**Author affiliations**
[1]The First Affiliated Hospital of Chongqing Medical University, Chongqing Key Laboratory of Ophthalmology, Chongqing Eye Institute, Chongqing Branch (Municipality Division) of National Clinical Research Center for Ocular Diseases, Chongqing, People's Republic of China
[2]Chongqing Medical University, Chongqing, People's Republic of China
[3]Chongqing General Hospital, Chongqing, People's Republic of China

**Acknowledgements**  We are indebted to KH and WW from the First Affiliated Hospital of Chongqing Medical University, Chongqing, China for their help in supervising this manuscript.

**Contributors**  YX and HC: conceptualisation, methodology, data analysis and writing-original draft preparation. KS, MD, NG, XY, JX and RH: data curation. SZ: supervision. TZ: validation. WW and KH: writing-reviewing and editing. Finaly, KH accepts full responsability for the work and/or the conduct of the study, has acces to the data, and controlled the decision to publish.

**Funding**  This work was supported by the National Natural Science Foundation of China (grant nos 81870650, 82000883, 81900885 and 81970832), the Project Foundation of Chongqing Science and Technology Commission of China (grant nos CSTC2021jscx-gksb-N0017, cstc2020jcyj-msxmX0829, CSTB2022NSCQ-MSX1561 and cstc2021jcyj-msxmX0967) and Chongqing Talent Plan 'Contract programme' (grant no. cstc2022ycjh-bgzxm0121).

**Competing interests**  None declared.

**Patient and public involvement**  Patients and/or the public were not involved in the design, or conduct, or reporting or dissemination plans of this research.

**Patient consent for publication**  Not applicable.

**Ethics approval**  This study involves human participant. The study adhered to the Declaration of Helsinki or relevant guidelines and regulations, and was approved by the ethical review committee of the First Affiliated Hospital of Chongqing Medical University, Chongqing, China, with the ethics approval reference number 2021-648. Participants gave informed consent to participate in the study before taking part.

**Provenance and peer review**  Not commissioned; externally peer reviewed.

**Data availability statement**  Data are available upon reasonable request.

**ORCID iD**
Ke Hu http://orcid.org/0000-0002-8055-389X

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
