## [Reviewer comments · BMJ Open]

ARTICLE DETAILS

TITLE (PROVISIONAL)	Myopia Prevalence and Ocular Biometry in Children and Adolescents at Different Altitudes: A Cross-sectional Study in Chongqing and Tibet, China
AUTHORS	Xiang, Yongguo; Cheng, Hong; Sun, Kexin; Zheng, Shijie; Du, Miaomiao; Gao, Ning; Zhang, Tong; Yang, Xin; Xia, Jiuyi; Huang, Rongxi; Wan, Wenjuan; Hu, Ke

VERSION 1 – REVIEW

REVIEWER	HUSSAINDEEN, JAMEEL RIZWANA SANKARA NETHRALAYA, BINOCULAR VISION CLINIC
REVIEW RETURNED	07-Oct-2023

GENERAL COMMENTS	In this Cohort study, the authors have tried to assess the impact of altitude on various parameters of Myopia in Chongqing and Tibet. The study is conducted well and presented clearly. Also most of the limitations are being presented explicitly. I have few concerns with respect to the unequal sample size and the influence of the same on the statistical outcomes and interpretations. Have the authors conducted Tests for normality? Especially when the Sample size is small in sub-group analysis, this can affect the power of the study. In Table 3, 4, the Sample size for the age groups must be mentioned. In figure 2, the SE seems to be on the higher side in Group 3. This is being reflected as higher Myopia prevalence in Group C (Table 5). Can the authors clarify this, and its relevance to the overall interpretation? It would be interesting and valuable to have details on the outdoor activities of these children to be able to correlate the UV exposure and its correlation with AL. I also assume that the other Myopia risk factors are matched in this sample to justify the association between altitude and AL. Can the authors comment about the same? How about Myopia progression in these children? Because having details of progression would also add value to the current observations. Overall the paper is well written, and have valuable findings in the pathophysiological understanding of Myopia.
--

REVIEWER	Zhong, Hua The First Affiliated Hospital of Kunming Medical University
REVIEW RETURNED	10-Oct-2023

GENERAL COMMENTS	This study investigated the differences in myopia prevalence and ocular biometry in children and adolescents from different altitude
--

	region in China. This school-based cross-sectional study reports comparable myopia prevalence in both region for students aged 6-8 and 9-11 years, but was lower and myopia progressed more slowly for students aged 12-14 and 15-18 years in Tibet, the higher-altitude region, than in Chongqing, the lower-altitude region. Two concerns: 1. Is there any differences on height, weight, and nutrition status etc. between the students in these two different altitude region? 2. The authors mentioned that the Tibet population had thinner cornea, lower IOP may be associated with long-term UVR exposure. Is there any data to show the UVR level in different regions? The association between UVR level/ exposure and ocular biometry could be analysed.
--	---

VERSION 1 – AUTHOR RESPONSE

Replies to Reviewer 1

1. Have the authors conducted Tests for normality? Especially when the Sample size is small in sub-group analysis, this can affect the power of the study.

Response: Thank you for your question regarding the normality test. Following your suggestion, we assessed the normality of the distribution of continuous variables using the one-sample Kolmogorov-Smirnov test and the histogram method of normal distribution. The independent samples t-test and analysis of variance (ANOVA) were employed to compare normally and approximately normally distributed continuous variables, and the Mann-Whitney U-test and Kruskal-Wallis H-test were used to compare heavily skewed continuous variables. According to the new statistical results, we made corresponding revisions in the figures, tables and results sections (Line 146-151, line 170-176, Table 3, Table 5, Figure S1b, and Figure S3b).

2. In Table 3, 4, the Sample size for the age groups must be mentioned.

Response: Thank you for your suggestion about labeling the sample size. In Table 3, the myopia prevalence section shows the sample size for each age group, which I have bolded. In addition, I have added sample size information in Table 4.

3. In figure 2, the SE seems to be on the higher side in Group 3. This is being reflected as higher Myopia prevalence in Group C (Table 5). Can the authors clarify this, and its relevance to the overall interpretation?

Response: As shown in Table 5, Group C exhibited smaller SE and higher prevalence of myopia, but the statistical analysis showed no statistical difference with Groups A and B. We believe that the possible reasons for this situation are as follows: first, Karo District and Drayab County (Group C) are more economically developed than Xiexiong Township (Group D) and Qucainka Township (Group B), therefore, the students in group C may have greater academic pressure and more exposure to electronic screens. Secondly, students in group C had relatively higher Km-values and greater anterior corneal surface astigmatism, which may have promoted higher myopia prevalence. Finally, due to the small sample size of the altitude subgroup, the overall situation may not be effectively reflected.

4. It would be interesting and valuable to have details on the outdoor activities of these children to be able to correlate the UV exposure and its correlation with AL. I also assume that the other Myopia risk factors are matched in this sample to justify the association between altitude and AL. Can the authors comment about the same?

Response: Thanks for raising this critical issue. Myopia is caused by both environmental and genetic risk factors. And the increasing prevalence of myopia can be largely explained by increased

educational pressures and reductions in the amount of time that children spend outdoors. Questionnaires were frequently used for evaluation of the time spent outdoors, but they represent a potential source of error due to inaccurate reporting or recall bias of participants. Wearable detectors can capture more objective data on time spent outdoors, but require more funding and observation time[1]. In the initial stages of study design, we indeed contemplated the inclusion of outdoor activity time as a significant variable. However, budget and time constraints prevented us from obtaining this data. In addition, I particularly agree with you that the relationship between altitude and AL can be evaluated more effectively by matching the two populations for other risk factors for myopia. However, we could not realize it in this study due to the lack of corresponding data. In the future studies, we are committed to incorporating risk factors like outdoor activity time to comprehensively assess their potential impact on study outcomes.

5.How about Myopia progression in these children? Because having details of progression would also add value to the current observations.

Response: We sincerely regret the absence of follow-up tracking. The time and budget limitations of the research project were the primary constraints preventing us from conducting a prolonged follow-up. In upcoming studies, we are committed to securing additional resources to facilitate a more comprehensive and in-depth investigation, including the follow-up of trial samples.

Replies to Reviewer 2

1.Is there any differences on height, weight, and nutrition status etc. between the students in these two different altitude region?

Response: Thanks for raising this critical issue. We acknowledge the importance of variables such as height, weight, and nutritional status to the study, and previous studies have reported body mass index and height as risk factors for myopia[2]. Regrettably, due to resource constraints and the study's scope, we did not conduct investigations into these factors in this particular study. However, through our observations in the survey, we found that most Tibetan children and adolescents showed poorer nutritional status, shorter heights and lighter weights compared to children and adolescents in Chongqing. Previous studies have reported similar findings.[3][4] Therefore, we believe that these indicators you suggest may be associated with the slower progression of myopia in Tibetan children and adolescents.

2.The authors mentioned that the Tibet population had thinner cornea, lower IOP may be associated with long-term UVR exposure. Is there any data to show the UVR level in different regions? The association between UVR level/ exposure and ocular biometry could be analysed.

Response: Previous studies have shown that the intensity of ultraviolet radiation increases by 10-12% for every 1,000 meters of elevation gain.[5] In addition, the average daily sunshine duration in Qamdo is 7.18 hours, which is about twice as long as the sunshine duration in Chongqing (3.54 hours) (<https://www.weather2visit.com/>). Thus, UVR levels in Qamdo are higher than in Chongqing, but unfortunately we were unable to obtain specific UVR exposures for both populations.

References

- [1].Xiong S, Sankaridurg P, Naduvilath T, Zang J, Zou H, Zhu J, Lv M, He X, Xu X. Time spent in outdoor activities in relation to myopia prevention and control: a meta-analysis and systematic review. *Acta Ophthalmol.* 2017 Sep;95(6):551-566.
- [2].Chen N, Sheng Y, Wang G, Liu J. Association between physical indicators and myopia in American adolescents: NHANES 1999-2008. *Am J Ophthalmol.* 2023 Dec 25:S0002-9394(23)00521-4.
- [3].Dang S, Yan H, Yamamoto S. High altitude and early childhood growth retardation: new evidence from Tibet. *Eur J Clin Nutr.* 2008 Mar;62(3):342-8.
- [4].Li X, Li Y, Xing X, Liu Y, Zhou Z, Liu S, Tian Y, Nima Q, Yin L, Yu B. Urban-rural disparities in the association between long-term exposure to high altitude and malnutrition among children under 5 years old: evidence from a cross-sectional study in Tibet. *Public Health Nutr.* 2022 Sep 13;26(4):1-10.
- [5].Narayanan DL, Saladi RN, Fox JL. Ultraviolet radiation and skin cancer. *Int J Dermatol.* 2010 Sep;49(9):978-86.

VERSION 2 – REVIEW

REVIEWER	Zhong, Hua The First Affiliated Hospital of Kunming Medical University
REVIEW RETURNED	10-Mar-2024
GENERAL COMMENTS	The Authors have thoroughly and extensively revised and amended the manuscript based on the comments from the Reviewers. I have no further comments or queries pertaining to this manuscript.

VERSION 2 – AUTHOR RESPONSE